# γ-Polyglutamic Acid Production, Biocontrol, and Stress Tolerance: Multifunction of *Bacillus subtilis* A-5 and the Complete Genome Analysis

**DOI:** 10.3390/ijerph19137630

**Published:** 2022-06-22

**Authors:** Naling Bai, Yu He, Hanlin Zhang, Xianqing Zheng, Rong Zeng, Yi Li, Shuangxi Li, Weiguang Lv

**Affiliations:** 1Eco-Environmental Protection Research Institute, Shanghai Academy of Agricultural Sciences, Shanghai 201403, China; bainaling@saas.sh.cn (N.B.); hyu0420@163.com (Y.H.); zhanghanlinchick@163.com (H.Z.); zxqfxf@163.com (X.Z.); zengrong@saas.sh.cn (R.Z.); 2Environment and Plant Protection Institute, Chinese Academy of Tropical Agricultural Sciences, Haikou 571101, China; wish.0310@163.com; 3Agricultural Environment and Farmland Conservation Experiment Station, Ministry Agriculture and Rural Affairs, Shanghai 201403, China; 4Key Laboratory of Low-Carbon Green Agriculture, Ministry of Agriculture and Rural Affairs, Shanghai 201403, China; 5Shanghai Key Laboratory of Horticultural Technology, Shanghai 201403, China

**Keywords:** *Bacillus subtilis*, genome analysis, γ-PGA production, pathogen antagonism, salinity/alkalinity tolerance

## Abstract

*Bacillus subtilis* A-5 has the capabilities of high-molecular-weight γ-PGA production, antagonism to plant pathogenic fungi, and salt/alkaline tolerance. This multifunctional bacterium has great potential for enhancing soil fertility and plant security in agricultural ecosystem. The genome size of *B. subtilis* A-5 was 4,190,775 bp, containing 1 Chr and 2 plasmids (pA and pB) with 43.37% guanine-cytosine content and 4605 coding sequences. The γ-PGA synthase gene cluster was predicted to consist of *pgsBCA* and factor (*pgsE*). The γ-PGA-degrading enzymes were mainly pgdS, GGT, and cwlO. Nine gene clusters producing secondary metabolite substances, namely, four unknown function gene clusters and five antibiotic synthesis gene clusters (surfactin, fengycin, bacillibactin, subtilosin_A, and bacilysin), were predicted in the genome of *B. subtilis* A-5 using antiSMASH. In addition, *B. subtilis* A-5 contained genes related to carbohydrate and protein decomposition, proline synthesis, pyruvate kinase, and stress-resistant proteins. This affords significant insights into the survival and application of *B. subtilis* A-5 in adverse agricultural environmental conditions.

## 1. Introduction

The excessive application of traditional chemical inputs (e.g., fertilizers and pesticides) in agricultural environments leads to a series of problems, including crop losses, environmental pollution, and pathogen resistance. The application of multifunctional microorganisms is of great importance to the synchronous reduction of chemical fertilizers and pesticides, soil restoration, food security, agricultural productivity, and ecological agriculture development [1]. Microorganisms capable of promoting plant growth as a new fertilizer, as well as controlling pathogens as a novel pesticide, have become an emerging research target worldwide as they have the potential to improve the scientific and technological innovation of green agricultural inputs [2].

*Bacillus* can produce many kinds of metabolites, such as amino acids, antibiotics, vitamins, nucleosides, sugars, and various enzymes. It, thus, acts as an important resource strain for the utilization of microbial metabolites [3]. *B. subtilis* has been widely applied in novel fertilizers, biological feeds, green pesticides, and other products, because it is the internationally recognized safe strain. The category, function, and related mechanisms of its metabolites have also been widely investigated [4]. The related functional genes are typically explored by routine methods in the following order: functional substance separation, purification, identification, gene mapping, cloning, and sequencing. In recent years, the rapid development of omics technology has provided a new and direct way to find novel genes, obtain the structural basis of genes, study the interactions between genes, and comprehensively clarify the molecular mechanisms of the biological function, expression, and evolution of a given strain.

Gamma-polyglutamic acid (γ-PGA) is an environmentally friendly plant growth stimulant with a molecular weight ranging from 10 to 10,000 kDa. As a biopolymeric and biodegradable substance, γ-PGA shows broad application prospects in various fields, such as agriculture, food, cosmetics, medicine, and waste treatment. The commercial production of γ-PGA is mainly attributed to *Bacillus* species, including wild type and modified strains. Multifunctional *Bacillus* strains, with the ability to produce γ-PGA and antagonize diseases, have been studied, e.g., *B. licheniformis* WX-02 (co-producing lichenysin and γ-PGA) [5] and *B. subtilis* NJtech489 (co-producing surfactin and γ-PGA) [1]. These strains facilitated the manufacture of multicomponent organic fertilizers with excellent fertilizer efficiency and simultaneous biocontrol function. Subsequently, the underlying molecular mechanisms of the multifunctional microbes were further analyzed. Jiang et al. [6] found that *B. velezensis* HG18, a biocontrol and growth-promoting strain at low temperature, possessed genes related to chitin degradation (6), glucanase (2), chitosanase (1), lipopeptide antibacterial substance (fengycin, surfactin) synthesis (2), and bacteriocin (subtilin, bacillolysin) synthesis (2), but had no γ-PGA synthesis gene clusters. The genome of *B. velezensis* 83 contained 10 secondary metabolite biosynthesis gene clusters related to biocontrol, along with genes related to γ-PGA and indole-3-acetic acid production, which would facilitate plant growth [7]. However, strain 83 was glutamate-independent in γ-PGA synthesis and the yield was only 1.4 g/L. In fact, these multifunction features do not commonly coexist in *Bacillus* species. *B. subtilis* 168, the model strain of *Bacillus*, can neither resist pathogens nor produce γ-PGA; *B. velezensis* FZB42 (formerly *B. amyloliquefaciens* FZB42), an effective biocontrol and commercially available agent, cannot synthesize γ-PGA, even with glutamate addition. The difference in native promoters most likely causes the variation of gene expression observed, although both strains 168 and FZB42 possesses the synthetase operon [8]. Researchers have also investigated the relationship between the production of antimicrobial substances and growth-promoting stimulants. Zhang et al. [9] noted that knockout of the genes encoding antimicrobial compounds (surfactin, iturin, fengycin, and bacillaene) was beneficial for increasing γ-PGA production. On the contrary, Qiu et al. [5] found that lichenysin promoted acetoin utilization, enhanced the metabolic flux of energy synthesis and tricarboxylic acid cycle, and thus facilitated γ-PGA polymerization in *B. licheniformis* WX-02.

*B. subtilis* A-5 can ferment γ-PGA with high molecular weight, and a pot experiment showed that γ-PGA increased fertilizer utilization efficiency and crop yield at a relatively low application rate (2.8–4.8 mg/kg) [10]. Additionally, strain A-5 antagonizes plant pathogenic fungi and confers tolerance to saline and alkaline stress, indicating its wide application potential in sustainable agricultural systems. However, it is difficult to fully understand all functional genes using traditional analysis methods. In view of this, our study aimed to use the combination of second- and third-generation sequencing technologies to sequence the whole genome of strain A-5. On this basis, functional genes involved in γ-PGA and antimicrobial compound synthesis and stress resistance were comprehensively investigated. The findings of this study lay the foundation for genetic engineering modification and multifunctional biofertilizer development.

## 2. Materials and Methods

### 2.1. Microbes Used in the Assay

*B. subtilis* A-5, isolated from homemade *natto*, can produce a high yield of γ-PGA, antagonize plant pathogens, and resist salinity and alkalinity. It was preserved at the China Center for Type Culture Collection (CCTCC No. 2019157). Plant pathogens (*Magnaporthe grisea*, *Rhizoctonia solani*, and *Fusarium oxysporum*) were preserved at the Eco-environmental Protection Research Institute, Shanghai Academy of Agricultural Sciences.

### 2.2. γ-PGA Fermentation Analysis of B. Subtilis A-5

*B. subtilis* A-5 was inoculated in Luria-Bertani (LB) medium at 37 °C and 200 rpm overnight. Then, 2% inocula were transferred to the optimized fermentation (FF) medium at 37 °C and 200 rpm for 48–96 h. The FF medium consisted of the following components (g/L): glucose, 40.0; sodium glutamate, 40.0; yeast extract, 5.0; MgSO_4_, 0.25; K_2_HPO_4_, 2.0. The pH was set to 7.0–7.2. The γ-PGA yield was measured using a cetyltrimethylammonium bromide-dependent spectrophotometer method and a UV-vis spectrophotometer (Puxi T6, Beijing, China). Its characteristics (number-average molecular weight, Mn; weight-average molecular weight, Mw; and polydispersity index, PDI) were measured by a gel permeation chromatography system (Shimadzu LC-20A, Kyoto, Japan) [10].

### 2.3. Analysis of Antagonistic Activity against Plant Pathogens

Plant pathogens were placed in the middle of a potato dextrose agar (PDA) medium plate using an Oxford cup (d = 0.80 cm) (Hopebio, Qingdao, China) and strain A-5 was inoculated 1.0 cm away from the plate edge. Plates only containing inoculated pathogen were used as the control treatment (CK). *B. velezensis* SS-20 was set as a positive inhibitor for comparison [11]. Furthermore, a glass slide was inserted into the PDA plate at a 45° angle to collect the growing mycelia. Treatments were performed in triplicate and cultured at 28 °C for 5–7 d. Mycelial morphology changes were viewed by an optics microscope (Olympus BX43, Center Valley, Japan) and a Cryo-scanning electron microscope (Cryo-SEM) (FEI Quanta 450, Hillsboro, OR, USA) after gradient ethanol dehydration. The inhibition rate = [1 − (mycelial diameter in experimental group/mycelial diameter in control group)] × 100% [12].

### 2.4. Salinity and Alkalinity Tolerance of B. subtilis A-5

To determine the salinity and alkalinity tolerance of *B. subtilis* A-5, the strain was inoculated in LB medium with a pH and a NaCl concentration ranging from 5.5 to 10.5 and 1% to 12% (*w*/*v*), respectively. Then, treatments were cultured at 37 °C for 2–4 d. Growth biomass was used as the criteria and measured spectrophotometrically at an optical density (OD) of 600 nm. Positive and negative control strains (*B. velezensis* SS-20 and *B. subtilis* FD4-1) were also included to compare the statistical significance. Furthermore, the tolerance of strain A-5 to salinity and alkalinity combined was determined by increasing the salt content and pH.

### 2.5. Whole Genome Sequencing and Analysis of B. subtilis A-5

After inoculation for 10–12 h (late logarithmic stage) in LB medium at 37 °C and 200 rpm, strain A-5 was collected to extract the genomic DNA using the Wizard^®^ Genomic DNA Purification Kit (Promega, Madison, WI, USA) according to the manufacture’s protocol. After purification, genomic DNA was quantified by a TBS-380 fluorometer (Turner, CA, USA). High-quality DNA (OD_260/280_ = 1.8–2.0, >20 μg) was used for further experiments.

Genomic DNA was sequenced using a combination of PacBio RSII single molecule real-time (SMRT) and Illumina HiSeq PE150 sequencing platforms. For Pacific Biosciences sequencing, an aliquot of 15 μg of DNA was spun in a Covaris g-TUBE (Covaris, Woburn, MA, USA) at 6000 rpm for 60 s by a centrifuge (Eppendorf 5424, Hauppauge, NY, USA). Then, DNA fragments were purified, end-repaired, and ligated with SMRTbell sequencing adapters (Pacific Biosciences, Menlo Park, CA, USA). The resulting sequencing library was purified three times with 0.45× the volume of Agencourt AMPure XP beads (Beckman Coulter Genomics, Newton, MA, USA). Next, a ~10 kb insert library was prepared and sequenced on one SMRT cell using standard methods. The Illumina data were used to evaluate and elevate genome completeness. The clean data reads were assembled into a contig using a hierarchical genome assembly process and canu [13]. The last circular step was checked and finished manually, thus generating a complete genome with seamless chromosomes and plasmids. Finally, error correction of the PacBio assembly results was performed using the Illumina reads provided by Pilon. Genome analysis was mainly carried out using the Majorbio Cloud Platform (https://cloud.majorbio.com, assessed on 3 March 2020). Glimmer [14], tRNA-scan-SE [15], and Barrnap were used for the prediction of coding sequence (CDS), tRNA, and rRNA, respectively. The predicted CDSs were annotated from non-redundant protein sequence (NR), Swiss-Prot, Pfam, gene ontology (GO), clusters of orthologous groups of proteins (COG), and Kyoto Encyclopedia of Genes and Genomes (KEGG) databases using alignment tools such as BLAST, Diamond, and HMMER. Briefly, each set of query proteins was aligned with the databases, and annotations of the best-matched subjects (E-value < 10^−5^) were obtained for gene annotation. The carbohydrate-active enzyme (CAZy) database was used to annotate carbohydrate active enzymes. AntiSMASH 6.0.1 was applied to analyze the secondary metabolite synthesis gene clusters, and TMHMM 2.0 and PSORTb 3.0.3 were used to predict the transmembrane helices in the proteins and subcellular locations of proteins, respectively.

### 2.6. Evolution and Comparative Genome Analysis of B. subtilis A-5

Thirty housekeeping genes (*dnaG*, *frr*, *infC*, *nusA*, *pgk*, *pyrG*, *rplA*, *rplB*, *rplC*, *rplD*, *rplE*, *rplF*, *rplK*, *rplL*, *rplM*, *rplN*, *rplP*, *rplS*, *rplT*, *rpmA*, *rpoB*, *rpsB*, *rpsC*, *rpsE*, *rpsI*, *rpsJ*, *rpsK*, *rpsM*, *smpB*, and *tsf*) were selected to construct the phylogenetic tree using MEGA 7.0 and the neighbor-joining method. The whole genome sequence of strain A-5 was compared with that of the model strain (*B. subtilis* 168, NC_000964.3), γ-PGA-producing strain (*B. subtilis* KH2, NZ_CP018184.1), and plant disease-resistant and growth-promoting bacteria (*B. subtilis* XF-1, NC_020244.1) using Mauve 2.3.1.

## 3. Results

### 3.1. γ-PGA Fermentation by B. subtilis A-5

*B. subtilis* A-5 yielded 34.21 g/L γ-PGA with an ultra-high molecular weight (Mw = 4700 kDa) in the FF medium at 48 h (Table 1). The yield and Mw of γ-PGA decreased to 24.58 g/L and 1106 kDa, respectively, when the cultured time reached 96 h, indicating that strain A-5 contained γ-PGA degradation-related enzymes [16]. PDI refers to the distribution breadth of a polymer’s molecular weight; the larger the PDI is, the wider the molecular weight distribution. The PDI value of polymers produced via step-growth polymerization is usually greater than 2.0 [17]. In this study, the PDI value of γ-PGA produced by strain A-5 varied from 1.860 to 2.663, verifying that this γ-PGA belonged to a mixture with certain molecular weight ranges formed via stepwise polymerization.

### 3.2. Antagonistic Activity against Fungal Plant Pathogens

The inhibition rates of *M. grisea*, *R. solani*, and *F. oxysporum* on mycelia growth were 77.93%, 22.09%, and 6.66%, respectively (Figure 1). Compared with the positive strain *B. velezensis* SS-20, strain A-5 demonstrated a significant decrease (32.05%; *p* < 0.01) in the inhibition efficiency on *R. solani*, which was not observed in the other fungal plant pathogens. The inhibitory efficiency of strain A-5 on *F. oxysporum* was considerably low compared to that of pathogens of rice (*M. grisea* and *R. solani*). Microscopy revealed the growth and morphological changes of *M. grisea* with or without strain A-5 treatment (Figure 2). The mycelia of *M. grisea* showed severe morphological abnormalities in the interaction zones in situ as compared with those of the control at the 100 μm scale of observation (Figure 2a,b). Optical microscope and Cryo-SEM identified healthy, dense, and branched mycelium in the control treatment, with no abnormality or disruption (Figure 2c,e); however, the hyphae of *M. grisea* were disrupted, flattened, and shriveled when challenged with *B. subtilis* A-5 (Figure 2d,f). The damage to the mycelium structure eventually inhibited further growth of *M. grisea*. Notably, γ-PGA presented no effect on pathogen growth whether it was obtained from *B. subtilis* A-5 or from commercial ones (data not shown).

### 3.3. Analysis of the Salinity and Alkalinity Tolerance of B. subtilis A-5

Compared to the positive strain (*B. velezensis* SS-20) and negative strain (*B. subtilis* FD4-1), *B. subtilis* A-5 conferred a high-level tolerance to salinity and alkalinity. Strain A-5 exhibited stress resistance under 12% NaCl or pH 10.5 condition, but the OD_600_ was only 0.15 at the 4th d (Figure 3). Under 1–10% contents of NaCl, strain A-5 grew well, with OD_600_ ranging from 0.44 to 1.45. The OD_600_ of strain A-5 exceeded 0.70 in acidic condition (pH = 5.5). As the alkalinity increased, the bacterial biomass increased to OD_600_ = 1.48 (pH = 7.0) and then significantly decreased (*p* < 0.05). Furthermore, strain A-5 grew well in LB medium with up to 8–10% NaCl and pH 8.5–9.5 (data not shown).

### 3.4. General Features of the B. subtilis A-5 Genome

The complete genome of *B. subtilis* A-5 consisted of one chromosome (Chr) and two plasmids (pA and pB) with a length of 4,190,775 bp, 43.37% guanine-cytosine (GC) content, and 4721 genes (Table 2). A total of 4605 CDSs, which make up 3,664,629 bp of the total length, accounted for 87.45% of the entire genome. For Chr, pA, and pB, the total nucleotide lengths were 4,120,646, 64,309, and 5820 bp, respectively, and their CDSs were 4512, 87, and 6, respectively. In addition, the genome encoded 86 tRNAs, 30 rRNAs, and 108 sRNAs. A circular plot of the genome (Figure 4) showed the number of bases, GC skew, GC content, and location of all annotated CDSs. A total of 345 virulence factor genes were identified in the *B. subtilis* A-5 genome. Virulence factors of bacterial pathogens (VFDB) database annotation showed that these were mainly offensive virulence factor, defensive virulence factor, nonspecific virulence factor, and regulation of virulence-associated genes (Appendix A). Twenty-nine types of drug-resistant genes (239 in number) were identified in the *B. subtilis* A-5 genome. The dominated resistant drugs were macrolide antibiotic, fluoroquinolone antibiotic, penicillin, tetracycline antibiotic, cephalosporin, aminoglycoside antibiotic, and cephamycin (gene number > 15) (Appendix A). The genome sequencing data of *B. subtilis* A-5 have been submitted to GenBank (PRJNA824974).

### 3.5. Functional Gene Annotation in B. subtilis A-5

In general, 4605, 3947, 3655, 3358, 3337, and 2291 genes were annotated in NR, Swiss-Prot, Pfam, COG, GO, and KEGG databases, respectively. A total of 3358 genes were annotated in COG database, accounting for 72.92% of the whole genome. There were 3333, 22, and 3 genes located in Chr, pA, and pB, respectively. The genes were mainly distributed in four categories: information storage and processing, metabolism, cellular processes/signaling, and poorly characterized. Notably, the “function unknown” type had the highest proportion (27.04%) (Figure 5), indicating that there is great potential to investigate new genes and functions in *B. subtilis* A-5. The number of genes related to amino acid transport and metabolism, carbohydrate transport and metabolism, transcription, replication-recombination and repair, and cell wall/membrane/envelope biogenesis all exceeded 200, i.e., 298, 260, 259, 227, and 200, respectively. Among all the 25 functions, 5 functions were not annotated, namely: RNA processing and modification, general function prediction only, extracellular structures, nuclear structure, and cytoskeleton related genes (Figure 5).

In *B. subtilis* A-5, 3337 genes were annotated in the GO database, accounting for 72.46% of the entire genome (Figure 6). There were 3296, 36, and 5 genes located in Chr, pA, and pB, respectively. Functional genes were distributed in biological process, molecular function, and cellular component categories. Genes related to the integrated component of membrane, plasma membrane, and cytoplasm exhibited the highest abundances among the sub-functions (i.e., 1001, 567, and 387 genes, respectively). Furthermore, integral component of membrane (68.29%) and DNA binding (21.95%) were the two functions that dominated in the plasmid. The reaction of γ-PGA biosynthesis is located in the cell membrane, and secondary metabolites also need to be transported and secreted into the extracellular matrix through the cell membrane [18]. Thus, it is necessary to explore in-depth the importance of membrane-associated proteins in *B. subtilis* A-5, especially those in the plasmid.

A total of 2291 genes were annotated to the KEGG pathway, accounting for 49.75% of the total genes, and were divided into 6 metabolic pathways (Figure 7). Accordingly, 2282, 8, and 1 genes were predicted to be located in Chr, pA, and pB, respectively. Most of genes (56.67%) were distributed in the metabolism, followed by environmental information processing (14.49%) and genetic information processing (8.29%). In environmental information processing, membrane transport was the main metabolic pathway and contained 179 gene annotations, indicating the importance of intracellular and extracellular communication.

### 3.6. Analysis of Carbohydrate Active Enzyme Annotation Using the CAZy Database

A total of 145 genes encoding proteins were analyzed by CAZy database: 54 glycoside hydrolases (GHs), 45 glycosyl transferases (GTs), 8 polysaccharide lyases (PLs), 29 carbohydrate esterases (CEs), 2 carbohydrate-binding modules (CBMs), and 7 auxiliary activities (AAs) (Table 3). γ-PGA is first identified in the extracellular capsule, where it combines with polysaccharides, and is also secreted into the extracellular matrix [19]; thus, PLs may be related to the synthesis or secretion of γ-PGA in *B. subtilis* A-5. Saccharides and proteins are the main components of the residues of plant pathogens in soil [1]. The genes encoding chitinase, lysozyme, peptidoglycan hydrolase, glycosidases, dextran, mannose, lactose, and galactose, were successfully annotated in *B. subtilis* A-5, which was also beneficial for the biocontrol mechanism analysis. In addition, genes encoding pyruvate kinase (*pyk*, A-5_gene 3131), glutamate synthesis complex enzyme (*proABC*, A-5_gene 2206-2208), catalase (*katE*, A-5_gene 1035/1484/4340/4383), and stress-resistant protein genes were also identified in the *B. subtilis* A-5 genome. The existence of these genes is suggested to be one of the underlying mechanisms of the salinity and alkalinity tolerance of *B. subtilis* A-5.

### 3.7. Evolution and Comparative Genome Analysis of B. subtilis A-5

Evolutionary analysis of 30 housekeeping genes and the phylogenetic tree subsequently constructed (Appendix A) verified that strain A-5 belongs to *Bacillus subtilis*. *B. subtilis* A-5 and KH2, γ-PGA-producing strains, contained two and one plasmids, in addition to the chromosome, respectively, while *B. subtilis* 168 and XF-1 consisted of only one chromosome (Table 4). Strain A-5 had the highest number of CDSs in the genome (4605). The genome of XF-1 was the smallest (4.06 Mb), but its GC content was the highest (43.90%) among the four strains (*B. subtilis* A-5, 168, KH2, and XF-1). In addition, collinearity analysis of the genome of strain A-5 was performed by comparing with that of *B. subtilis* 168, *B. subtilis* KH2 [20], and *B. subtilis* XF-1 [21]. High similarity was observed in the genomes of the four *B. subtilis* strains (Figure 8). However, they significantly differed in terms of sequence insertion, turnover, translocation, and deletion.

### 3.8. Analysis of Genes Related to γ-PGA Metabolism

#### 3.8.1. Characteristics of Genes Related to γ-PGA Synthesis

In *B. subtilis* A-5, *pgsBCA* and the required factor (*pgsE*) were essential for γ-PGA synthesis (Figure 9), which functioned as a synthase complex. The transmembrane and subcellular localization of related proteins were predicted by TMHMM and PSORTb. *pgsB* (gene 4031) possessed a full length of 1182 bp nucleotides encoding 393 bp of amino acids (aa), which could catalyze γ-PGA synthesis intracellularly as a non-transmembrane protein by anchoring to the plasma membrane through an anchor hook. *pgsC* (gene 4030) contained 450 bp nucleotides encoding 149 aa; pgsC was predicated to have four transmembrane helixes and to be fixed on the plasma membrane to connect pgsB and pgsA. Protein pgsA (gene 4029) was predicted to contain one transmembrane helix in the 380 aa and to be anchored to the plasma membrane through the N-terminal transmembrane region and anchor hooks. Therefore, pgsA is also responsible for the extracellular transport of γ-PGA. *pgsE* (gene 4028) was annotated to encode the factor required for γ-PGA synthesis and one transmembrane helix was predicted in the membrane with 27–55 aa.

#### 3.8.2. Characteristics of Genes Related to γ-PGA Decomposition

Enzymes related to γ-PGA decomposition play vital roles in regulating the yield and molecular weight of γ-PGA. Studies have mainly focused on such enzymes as endo-type γ-PGA hydrolase (pgdS), γ-glutamyl transpeptidase (GGT), and DL-endopeptidases (cwlO) [22]. γ-PGA can be internally hydrolyzed into small peptides by pgdS and then externally hydrolyzed into glutamate monomers by GGT. cwlO plays an important role in γ-PGA production and degradation, and also affects its molecular size [23]. Genes encoding these γ-PGA-degrading enzymes, namely gene 4027 (*pgdS*, EC: 3.4.19.9), gene 2200/gene 4053 (*ggt*, EC: 2.3.2.2), and gene 3896 (*cwlO*, EC: 3.4.--), were all found in the genome of *B. subtilis* A-5 (Table 5).

Subcellular location analysis showed that pgdS and cwlO were both located in the cell wall with probabilities of 92.00% and 92.10%, respectively (Table 5), which is consistent with their functions. pgdS contained a strong transmembrane region (N-terminal) predicted by TMHMM. The nucleotide similarities between *pgdS* and *B. subtilis* KH2 (NZ_CP018184.1), *B. subtilis* XF-1 (NC_020244.1), and *B. subtilis* 168 (NC_000964.3) were 100%, 98.31%, and 98.90%, respectively. GGT (*ggt*, gene 2200) was an extracellular enzyme with a probability of 99.80% and a strong transmembrane helix (N-terminal signal peptide). This conclusion is consistent with the result of Wang et al. [24]. No transmembrane helix was predicted in cwlO (gene 3896). It presented 100%, 98.51%, and 97.82% similarities with *B. subtilis* KH2 (NZ_CP018184.1), *B. subtilis* XF-1 (NC_020244.1), and *B. subtilis* 168 (NC_000964.3), respectively. GGT was predicted to be located in the extracellular matrix (Table 5); the result was in coincidence with its role as an exo-hydrolase to hydrolyze those low molecular weight γ-PGA. Nucleotide similarities of *ggt* (gene 2200) with *B. subtilis* KH2 (NZ_CP018184.1), *B. subtilis* XF-1 (NC_020244.1), and *B. subtilis* 168 (NC_000964.3) were 100%, 97.85%, and 99.04%, respectively. Furthermore, another *ggt* (gene 4053) was also annotated in the genome, which was predicted to be an extracellular enzyme (97.20%) via PSORTb analysis. Gene 4053 showed only 29.04% similarity with gene 2200, indicating a low homology and a different category. After analysis using the UniProt database, it showed that the protein encoded by gene 4053 had 98.1% similarity with glutathione hydrolase-like YwrD proenzyme (O01218) in *B. subtilis* 168 and belonged to the glutamyl transferase family. Therefore, the probable role of GGT heteroisozymes (gene 4053) in γ-PGA degradation needs further investigation.

### 3.9. Analysis of the Secondary Metabolite Gene Clusters

Secondary metabolites synthesized by *Bacillus* have good antimicrobial properties, which can directly or indirectly stimulate plant systemic resistance and avoid pathogenic microbes [21]. In this study, nine secondary metabolite gene clusters were predicted and located in the Chr of *B. subtilis* A-5 by antiSMASH analysis. Surfactin, fengycin, bacillibactin, subtilosin_A, and bacilysin were annotated and are listed in Table 6. Four gene clusters (Cluster2, 4, 5, and 9) were predicted with unknown functions, including two terpenes, one T3PKS and one bacteriocin, suggesting that *B. subtilis* A-5 might synthesize new antimicrobial substances. Strain A-5 may have the potential for development and application in the agriculture and bio-pharmaceutical industries. All four strains (*B. subtilis* A-5, 168, KH2, and XF-1) contained antimicrobial gene clusters (e.g., surfactin, fengycin, bacillibactin, subtilosin, and bacilysin). Fourteen secondary antimicrobial gene clusters were annotated in *B. subtilis* 168; among them, bacillaene, sublancin 168, and thailanstatin A were only identified in strain 168.

## 4. Discussion

Excessive application of chemical fertilizers and pesticides causes serious environmental pollution [10]. Green agricultural input, characterized by their safety, eco-friendliness, and high efficiency, has become a research hotspot. Among them, multifunctional microorganisms have gained much attention because of their synergy with fertilizers and biocontrol [5,6]. *Bacillus* has been widely studied in terms of its roles in plant growth stimulant production, biological control, pollutant degradation, and the synthesis of degradable biomaterials [25].

In this study, the molecular weight of γ-PGA changed dynamically when fermented in the FF medium. The maximum Mw was 4700 kDa, which then gradually decreased to 1106 kDa with a cultivation time of 96 h (Table 1). During polymerization, glutamic acid was gradually added to the tail of γ-PGA_n-1_ for longer γ-PGA synthesis, thus leading to viscosity increase and dissolved oxygen decrease in the fermentation system. Continuous consumption of nutrients in the culture medium and the presence of γ-PGA degradation-related enzymes (pgdS, GGT, and cwlO) jointly affected the ultimate yield and molecular weight of γ-PGA [26]. Additionally, *B. subtilis* A-5 inhibited the mycelial growth of *M. grisea*, *R. solani*, and *F. oxysporum* (6.66–77.93%); it also conferred a high tolerance of salinity and alkalinity (NaCl 1–10% and pH 5.5–9.5) (Figure 1, Figure 2 and Figure 3). Therefore, it can be hypothesized that *B. subtilis* A-5 has potential as a valuable bio-organic fertilizer.

The functional genes related to γ-PGA metabolism, pathogen control, and salt/alkaline stress resistance have been successfully explored in the whole genome of *B. subtilis* A-5. Annotation by the COG database showed that, with the exception of the unknown functions, amino acid transport and metabolism were the main functions, which explained the production and secretion of γ-PGA and secondary substances in *B. subtilis* A-5. Similarly, in the genome of *B. licheniformis* WX-02, a γ-PGA producer, the protein functions were significantly enriched in unknown functions, carbohydrate transport/metabolism, and replication/recombination/repair [27]. The main functions annotated by KEGG were carbohydrate metabolism, global and overview maps, and amino acid metabolism, which was similar to the metabolic pathways analyzed in the γ-PGA-synthesizing strain *B. amyloliquefaciens* LL3 [28]. The proteins related to carbohydrate and protein decomposition, proline synthesis, and stress resistance in *B. subtilis* A-5 may be the contributory factors to biocontrol and stress resistance. For example, proline has strong hydration ability to maintain osmotic balance, prevent cell deformation, and remove reactive oxygen species; *Bacillus* can accumulate proline by de novo synthesis to cope with osmotic changes and ensure survival and growth [29]. The dominating functions annotated in COG database were predicted to be “FUNCTION UNKNOWN” in pA and pB, suggesting that the unknown functional genes in *B. subtilis* A-5 may participate in important biological processes [30] and are thus worthy of further research.

The γ-PGA synthase complex was composed of pgsBCA and factor (pgsE), where pgsB was the catalytic component and could provide ATP for catalysis. pgsC formed a catalytic site with pgsB and then polymerized the substrate (glutamate acid). pgsA, located on the cell surface with a membrane-anchoring region, assisted γ-PGA transportation and played an important role in γ-PGA prolongation [31]. The pgsBCA complexes are different to a certain degree in the glutamate-dependent/independent strains, especially pgsA [32], since it is vital for γ-PGA yield and molecular weight. Enzyme cwlO was the only peptidoglycan hydrolase with extracellular activity and was not regulated by γ-PGA degradation system [23]. In theory, knockout of pgdS, GGT, or cwlO may shut down the γ-PGA degradation pathway, thus facilitating the expression of γ-PGA synthase cluster and product accumulation. For example, after knockout of the degradation enzymes, γ-PGA production of strain NK-E11 significantly increased from 4.15 g/L to 9.18 g/L in a previous study [33]. However, in contrast, deletion of *pgdS* and *ggt* inhibited and increased γ-PGA production, respectively, in *B. licheniformis* RK14-46 [24]. Feng et al. [22] found that the γ-PGA yield decreased to only 2.69 g/L when *pgdS*, *GGT*, and *cwlO* were synchronously knocked out. Generally, these results indicate the different intrinsic interactions of pgdS, GGT, and cwlO in γ-PGA production and decomposition. Metal ions, such as Fe^3+^, K^+^, and Mn^2+^ have also been reported to influence synthesis and accumulation of γ-PGA [34,35]. The yield and molecular weight distribution of γ-PGA can be regulated by enhancing *pgsBCA* expression, knocking out γ-PGA-degrading enzymes, and regulating the content of elements in fermentation system.

Whole genome analysis revealed that *B. subtilis* A-5 also possessed the genes encoding secondary antimicrobial substances (Table 6). Surfactin is a cyclic lipopeptide biosurfactant synthesized by the non-ribosomal peptide synthetase (NRPS) pathway, which can reduce the surface tension of water from 72 mN/m to 27 mN/m at 20 μm. It has various biological functions, such as hemolysis, antibiosis, anti-mycoplasma/chlamydia, antiviral action, and tumor cell proliferation inhibition [36]. Fengycin is another cyclic lipopeptide antibiotic produced by *Bacillus*, which consists of 10 aa residues and 1 β-hydroxy fatty acid. It exerts antibacterial effects by disrupting the fungal cell membrane and damaging its structure [37]. Chung et al. [38] found that fengycin inhibited *Staphylococcus aureus* colonization by inhibiting quorum sensing. Bacillibactin is a kind of catechol siderophore with high affinity to competitively bind to the soluble irons necessary for the growth and activity of pathogens. Subtilosin_A is another uncommon lantibiotic with a special structure composed of 35 aa, which has inhibitory activity to both gram-positive and gram-negative bacteria. Bacilysin, a peptide antibacterial substance, belongs to the ribosomal synthesis pathway. It has a wide spectrum against bacteria and strong inhibitory effects on some fungi (e.g., *Candida albicans*) [39]. Furthermore, alkaline serine protease (gene 2005) was found in *B. subtilis* A-5 with a similarity rate of 99.4% (data not shown), which may reduce root-knot nematodes [40]. Therefore, strain A-5 may achieve antimicrobial effects through the combined actions of lipopeptide antibiotics, functional proteins, and competition effects. As for the internal relationship between γ-PGA metabolism, biocontrol efficiency, and stress tolerance, Wang et al. [41] concluded that *pgsB* deletion impaired the bacterial colonizing capacity and lowered the biocontrol rate from 62.08% to 14.22% against *Fusarium* root rot. They also found that knockout of *pgdS* and *ggt* led to a considerable improvement in biocontrol efficiency. Furthermore, the expression level of stress-related genes was notably changed by *pgsB* deletion [41], demonstrating the importance of the interconnectedness of γ-PGA production, biocontrol, and stress tolerance. Based on the linkage of amide bonds between α-NH_2_ and γ-COOH groups in the γ-PGA backbone and polyanionic characteristics, it may adsorb cations to form dissolved complexes for slow release. Thus, the presence of γ-PGA may facilitate to alleviate salt/alkaline stress [42]. In the future, the mechanism of the joint effect should be studied in-depth and a field-scale investigation and verification need to be carried out.

## 5. Conclusions

The multifunctional bacterium *B. subtilis* A-5 can produce γ-PGA (maximum Mw: 4700 kDa), antagonize plant fungal pathogens (6.66–77.93%), and tolerate salinity and alkalinity stress (NaCl 1–10% and pH 5.5–9.5). The genome size of *B. subtilis* A-5 was 4,190,775 bp, containing 1 Chr and 2 plasmids (pA and pB) with 43.37% GC content and 4605 CDSs. The γ-PGA synthase gene cluster, *pgsBCA* and factor (*pgsE*), were annotated and the enzyme complex was predicted to work in the cell membrane. γ-PGA-degrading related enzymes were mainly pgdS, GGT, and cwlO, located in the cell wall or extracellular matrix. Nine secondary metabolite gene clusters, including four unknown function gene clusters and five antibiotic synthesis gene clusters (surfactin, fengycin, bacillibactin, subtilosin_A, and bacilysin), were predicted in the genome of *B. subtilis* A-5. In addition, *B. subtilis* A-5 contained genes related to carbohydrate and protein decomposition, proline synthesis, pyruvate kinase, and stress-resistant proteins, which could facilitate biocontrol and stress resistance. In conclusion, *B. subtilis* A-5 has great potential for application in sustainable agriculture systems. This study provides theoretical support to the investigation of the underlying metabolic mechanism of *B. subtilis* A-5 and impetus to construct engineered strains. In the future, the interior mechanism of the different effects of this strain should be studied and field-scale application should be carried out.

## Figures and Tables

**Figure 1 ijerph-19-07630-f001:**
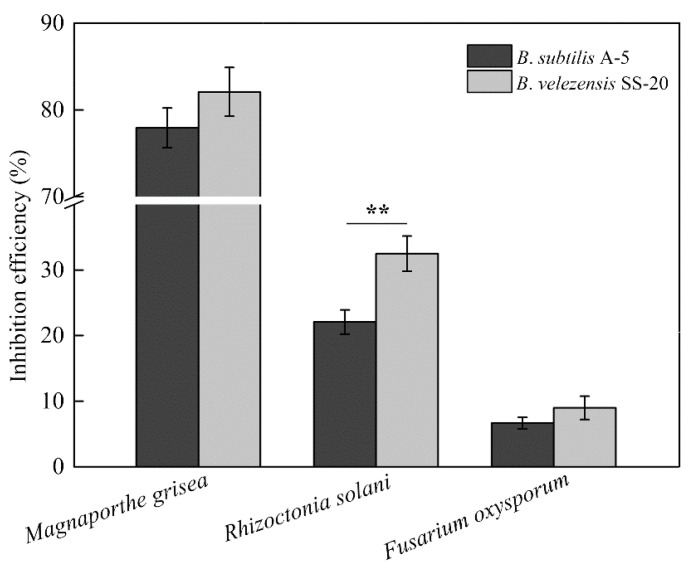
Analysis of the antagonistic efficiency of *B. subtilis* A-5 against fungal plant pathogens. “**” indicates statistically significant difference at the 0.01 level (*p* < 0.01).

**Figure 2 ijerph-19-07630-f002:**
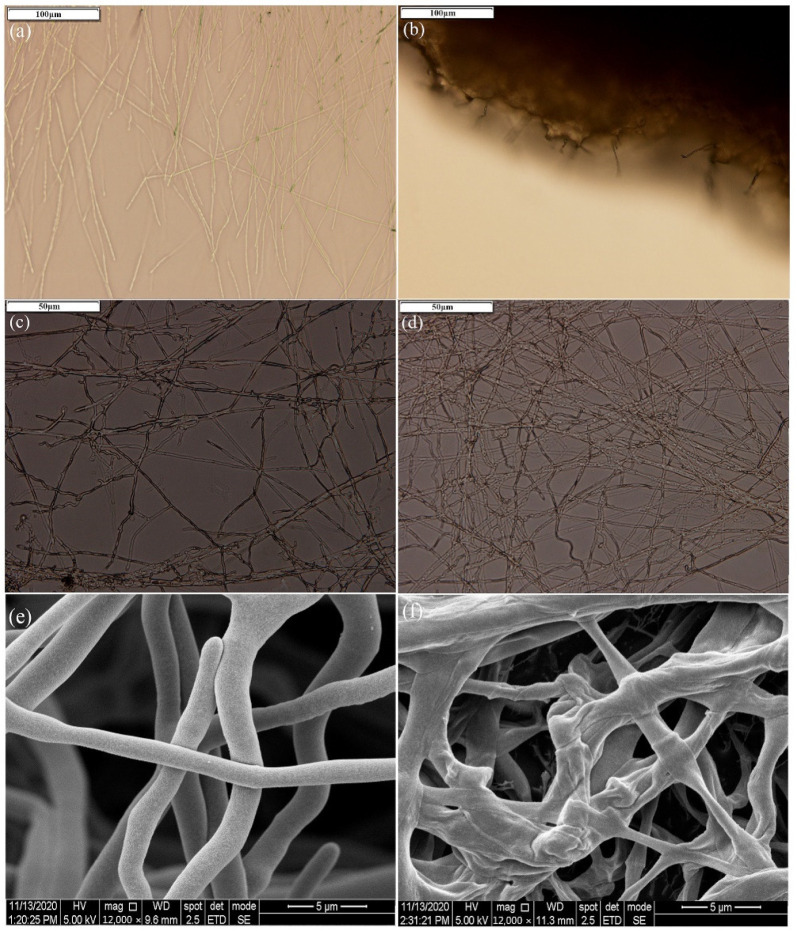
Effects of *B. subtilis* A-5 on mycelial growth and the morphological characteristics of *M. grisea* as observed using Olympus BX43 microscope (**a**–**d**) and Cryo-SEM (**e**,**f**). (**a**,**c**,**e**) refer to CK samples; (**b**,**d**,**f**) refer to fungal mycelia inhibited by *B. subtilis* A-5.

**Figure 3 ijerph-19-07630-f003:**
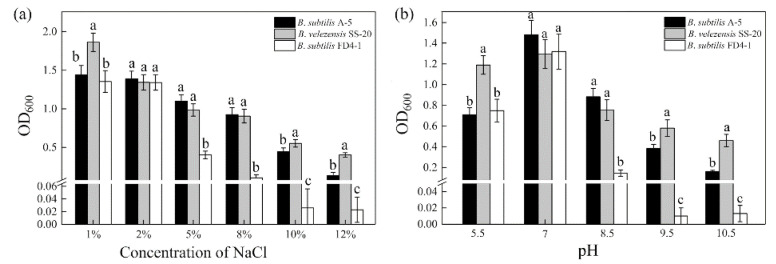
Salinity (**a**) and alkalinity (**b**) tolerance of *B. subtilis* A-5. The lowercase letters indicate significant differences among different strains (*p* < 0.05).

**Figure 4 ijerph-19-07630-f004:**
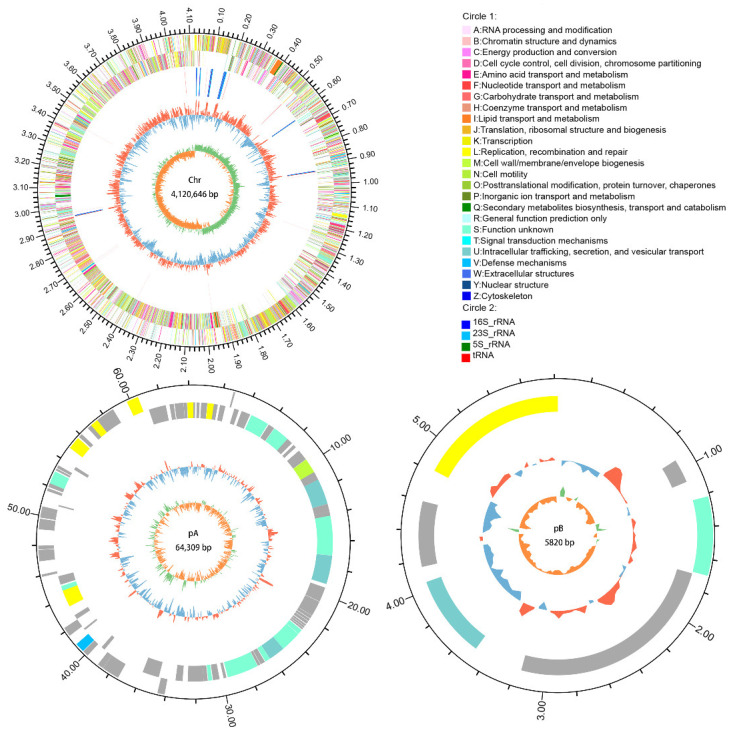
Genome map of *B. subtilis* A-5 (Chr, pA, and pB). Rings from the outside as follows: (1) scale marks, (2) CDSs information in the forward strand, (3) CDSs information in the reverse strand, (4) rRNA and tRNA genes, (5) GC content (deviation from average), (6) GC skew.

**Figure 5 ijerph-19-07630-f005:**
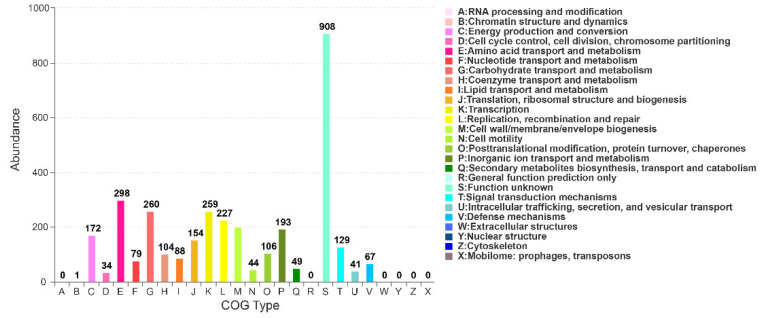
COG annotation classification of the *B. subtilis* A-5 genome.

**Figure 6 ijerph-19-07630-f006:**
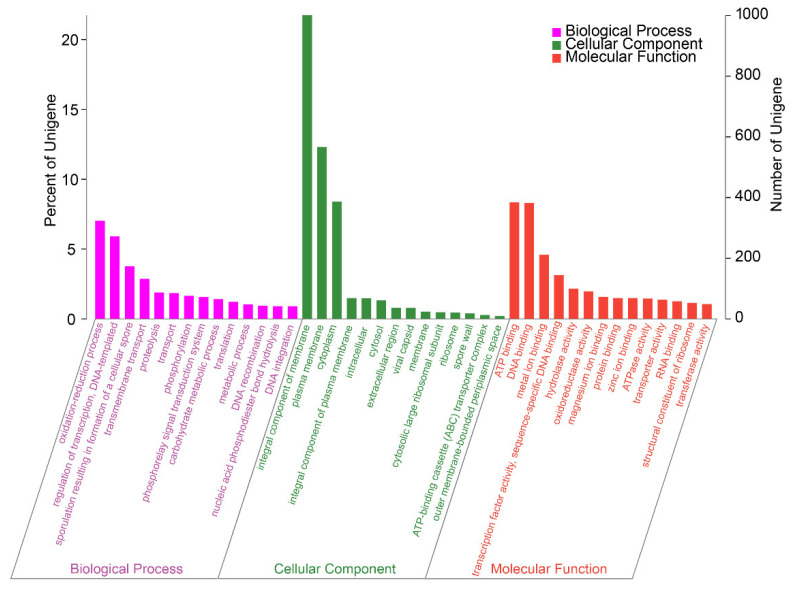
GO annotation classification of the *B. subtilis* A-5 genome.

**Figure 7 ijerph-19-07630-f007:**
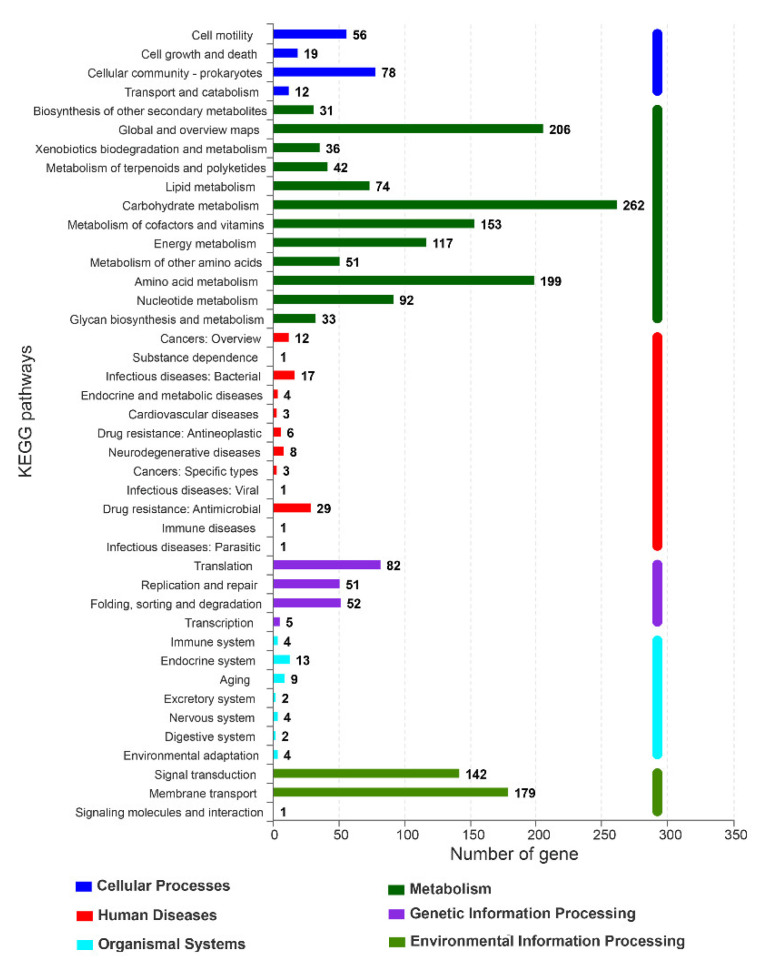
KEGG annotation classification of the *B. subtilis* A-5 genome.

**Figure 8 ijerph-19-07630-f008:**
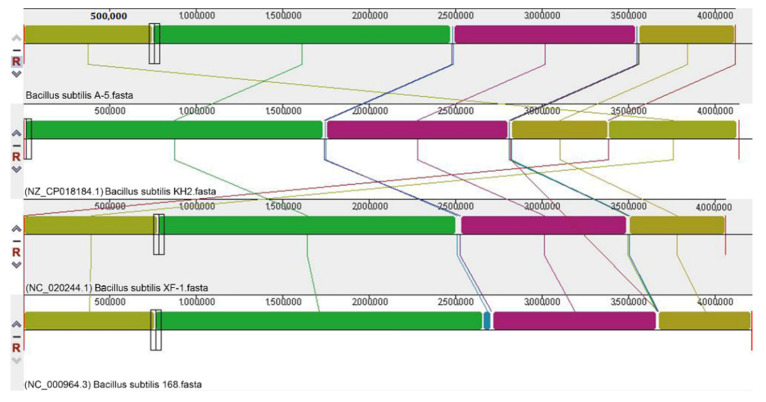
Synteny analysis of *B. subtilis* A-5, 168, KH2, and XF-1 by Mauve.

**Figure 9 ijerph-19-07630-f009:**
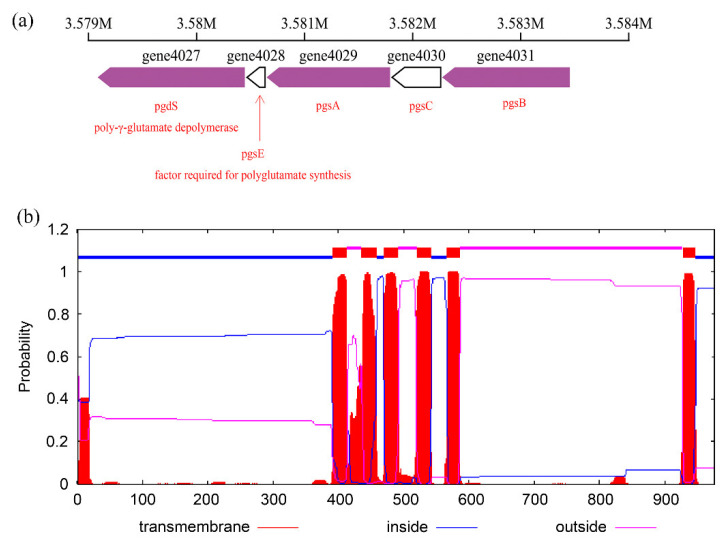
Map of the γ-PGA synthetase gene cluster (*pgsBCAE*) and the endo-type γ-PGA hydrolase gene (*pgdS*) in *B. subtilis* A-5 (**a**); analysis of the transmembrane structure of the γ-PGA synthase cluster by TMHMM (**b**).

**Table 1 ijerph-19-07630-t001:** Characteristics of γ-PGA fermented by *B. subtilis* A-5.

Fermentation Time (h)	γ-PGA Yield (g/L)	Retention Time (min)	Mn (kDa)	Mw (kDa)	PDI
48	34.21	18.933	1764	4700	2.663
96	25.58	17.070	596	1106	1.860

Note: γ-PGA, gamma-polyglutamic acid; Mn, number-average molecular weight; Mw, weight-average molecular weight; PDI, polydispersity coefficient.

**Table 2 ijerph-19-07630-t002:** Whole genome assembly of *B. subtilis* A-5.

Assembly	Length (bp)	GC Content (%)	Gene Number	CDSs	Bases Statistics
Chr	4,120,646	43.47	4628	4512	A:1,162,420; T:1,167,103; G:894,231; C:896,892
pA	64,309	37.66	87	87	A:19,279; T:20,809; G:10,822; C:13,399
pB	5820	40.74	6	6	A:1640; T:1809; G:960; C:1411

Note: Chr, chromosome; pA, plasmid A; pB, plasmid B; GC content, guanine-cytosine content; CDSs, coding sequences. A, T, G and C refer to adenosine, thymine, guanine, and cytosine, respectively.

**Table 3 ijerph-19-07630-t003:** CAZyme statistics for the *B. subtilis* A-5 genome.

Class Definition	Gene Number	Percentage of Total Genes (%)
AAs	7	4.83
CBMs	2	1.38
CEs	29	20.00
GHs	54	37.24
GTs	45	31.03
PLs	8	5.52

Note: AAs, CBMs, CEs, GHs, GTs and PLs refer to auxiliary activities, carbohydrate-binding modules, carbohydrate esterases, glycoside hydrolases, glycosyl transferases, and polysaccharide lyases, respectively.

**Table 4 ijerph-19-07630-t004:** Comparison of the characteristics of the whole genome sequences of *B. subtilis* A-5, 168, KH2, and XF-1.

Characteristic	*B. subtilis* A-5	*B. subtilis* 168	*B. subtilis* KH2	*B. subtilis* XF-1
CDSs	4605	4328	4377	4175
Total size (bp)	4,190,775	4,215,606	4,212,430	4,061,186
GC (%)	43.37	43.51	43.38	43.90
Antimicrobial secondary metabolites	9	14	9	10
Chromosome number	1	1	1	1
Plasmid number	2	0	1	0

Note: CDSs, coding sequences; GC, guanine-cytosine content.

**Table 5 ijerph-19-07630-t005:** Subcellular location analysis of γ-PGA degrading enzymes by PSORTb.

Subcellular Location	Prediction Probability (%)			
	pgdS (Gene 4027)	cwlO (Gene 3896)	GGT (Gene 2200)	GGT (Gene 4053)
Cytoplasmic	0.10	0.00	0.00	0.10
Cytoplasmic membrane	0.10	0.10	0.00	0.90
Cell wall	92.00	92.10	0.20	1.80
Extracellular	7.80	7.80	99.80	97.20

Note: pgdS, endo-type γ-PGA hydrolase; cwlO, DL-endopeptidases; GGT, γ-glutamyl transpeptidase.

**Table 6 ijerph-19-07630-t006:** Gene cluster analysis of the secondary metabolites of *B. subtilis* A-5.

Cluster ID	Type	Similar Cluster	Similarity (%)	Gene Number	Gene Length (bp)
Cluster1	NRPS	Surfactin	82	51	65,392
Cluster2	Terpene	-	-	25	20,807
Cluster3	NRPS	Fengycin	100	46	56,716
Cluster4	Terpene	-	-	22	21,899
Cluster5	T3PKS	-	-	48	41,096
Cluster6	NRPS	Bacillibactin	100	44	49,740
Cluster7	Sactipeptide-head_to_tail	Subtilosin_A	100	20	21,612
Cluster8	other	Bacilysin	100	45	41,417
Cluster9	Bacteriocin	-	-	17	14,546

Note: NRPS, non-ribosomal peptide synthetase; T3PKS, type III polyketide synthase. “-” refers to no annotation.

## Data Availability

The datasets used and/or analyzed during the current study are available in the GenBank (PRJNA824974).

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
