# Peer review of "γ-Polyglutamic Acid Production, Biocontrol, and Stress Tolerance: Multifunction of Bacillus subtilis A-5 and the Complete Genome Analysis"

_ijerph, 2022, doi:10.3390/ijerph19137630_

Round 1

Reviewer 1 Report

The MS by Bai, N., et al. was aimed to characterize the strain B. subtilis A-5. Authors showed γ-PGA production, antagonistic effect on plant pathogens and saline/pH tolerance. In addition, they obtained and analyzed the whole genome of this strains. I have some points that must be addressed:

·      Characterization of B. subtilis A-5 of antagonism of plant pathogens and saline/pH tolerance must be assayed using control with other Bacillus strains. Are these features unique in B. subtilis A-5 or they are common in other strains? Perhaps authors could use strains that not produce γ-PGA as those mentioned in introduction section (B. velezensis HG18, B. subtilis 168, B. amyloliquefaciens FZB42).

·      Authors found two plasmids, however they were ignored in the MS. COG (figure 5), GO (Figure 6) and KEGG (figure 7) annotations should specify the genes located in plasmids. Also, discussion section should mention about these plasmids. Does exist any advantages of metabolism, antimicrobial, soil adaptation, etc?      

·      KEEG annotation shows 262 genes involved in carbohydrates metabolism, however 145 genes were analyzed by CAZy database, What happened with the other ones?

·      Check line 170, the subtitle has punctuation error.

·      Font size in figures is small, especially in figures 2 and 4. It´s really hard to appreciate the inset letter in panels of figure 2. In figure 4, the COG function classification and other features of the genome is impossible to read, in addition if plasmid and chromosome have the same features, I think one code legend could explain the three circles.

·      Table 5. I think that the concept of Probability is ambiguous. Perhaps authors could explain this concept with a foot note.

·      Line 471, the scientific name must be written in italics.   

Reviewer 2 Report

The article deals with the analysis of an isolate B. subtilis A5 whole genome was sequenced and a search for genes with the capacity to confer properties to this bacterium as a biofertilizer and control agent of some phytopathogenic bacteria was established. The article is well written, well founded and the analysis and comparison of the complete genome of this strain with other organisms is excellent.

1. However, it is necessary that the authors not only focus on making a descriptive analysis of the genes found and that they coincide with the characteristics that they were looking for, but also mark the differences at the gene level at a more global level between the different strains compared.

 2.- The section where were evaluated the detrimental effect on the three phytopathogenic organism Magnaporthe grisea, Rhizoctonia solani, and Fusarium oxysporum, is important, but the authors need to include the comparison with a control that has already been reported as an inhibitor of these strains.

3.  In salinity studies, it is also  important a positive and negative control are also included, in order to be able to compare the results and their statistical significance.

Round 2

Reviewer 1 Report

The authors improved the manuscript and adequately addressed my concerns. I recommend the manuscript for publication.